# Microglia in Neurodegenerative Events—An Initiator or a Significant Other?

**DOI:** 10.3390/ijms22115818

**Published:** 2021-05-29

**Authors:** Gaylia Jean Harry

**Affiliations:** Molecular Toxicology Branch, Division of the National Toxicology Program, National Institute of Environmental Health Sciences, National Institutes of Health, MD E1-07, 111 T.W. Alexander Drive, Research Triangle Park, NC 27709, USA; harry@niehs.nih.gov

**Keywords:** microglia, neurodegeneration, neuroinflammation

## Abstract

A change in microglia structure, signaling, or function is commonly associated with neurodegeneration. This is evident in the patient population, animal models, and targeted in vitro assays. While there is a clear association, it is not evident that microglia serve as an initiator of neurodegeneration. Rather, the dynamics imply a close interaction between the various cell types and structures in the brain that orchestrate the injury and repair responses. Communication between microglia and neurons contributes to the physiological phenotype of microglia maintaining cells in a surveillance state and allows the cells to respond to events occurring in their environment. Interactions between microglia and astrocytes is not as well characterized, nor are interactions with other members of the neurovascular unit; however, given the influence of systemic factors on neuroinflammation and disease progression, such interactions likely represent significant contributes to any neurodegenerative process. In addition, they offer multiple target sites/processes by which environmental exposures could contribute to neurodegenerative disease. Thus, microglia at least play a role as a significant other with an equal partnership; however, claiming a role as an initiator of neurodegeneration remains somewhat controversial.

## 1. Introduction

The increasing evidence in support of the importance of the immune system in neurological disorders and degenerative processes has often led to the speculation that inflammatory factors and the associated neural cells play a role in the disease initiation process. While there is strong evidence for an association between the two, assigning a cause and effect is much more difficult and questionable. The majority of studies showing a beneficial effect of downregulating the neuroimmune response address roles of a contributory nature rather than an initiation process. In many of the associated disorders and degenerations, multiple confounding factors come into play that may bias those efforts to assign a role as an initiation factor. Major factors include the correlation between neurodegenerative diseases and increased age of the patient; subtle damage to the cerebrovascular system and decreased function of the blood brain barrier, (BBB) allowing for entry of blood-borne immune cells; and systemic changes that can influence disease processes, such as changes in the microbiome, systemic inflammatory responses, and hormonal levels. Within such complex disease processes, it is difficult to identify the initiating factor. In fact, there is limited understanding of the underlying “cause” of many of these more common diseases.

Identification of various cellular and structural features of the nervous system as contributors to neuroimmune responses has served to expand and enhance our appreciation, and maybe understanding, of the complexity of the responses that contribute to neurological and neurodegenerative-related disorders. As the primary central nervous system (CNS) immune cell, microglia perform critical functions in maintaining a healthy environment and respond to changes that occur in neurodegenerative diseases and neural injury [1,2]. While the presence of reactive and activated microglia has been observed in numerous animal models of human neurodegenerative disease as well as in patients, the overall evidence suggests that, in the context of a pathogen-free environment, a relatively benign innate immune response is evoked. However, how, and if, this environment can be altered or stressed in a manner to change this balance remains a critical question. With regards to the question of a role for neuroinflammation in the initiation of neurotoxicity, whether as a biological process or in response to an environmental or pharmacological factor, determining the actual contributory nature is of importance for regulating the progression of the disease process.

## 2. Microglia Morphology

Microglia are morphologically heterogeneous cells which comprise approximately 20% of the total cells in the brain. Under normal conditions, microglia assume a neural specific phenotype due to the CNS environment and retain a surveillance phenotype with fine processes extending into the surrounding microenvironment. This allows the cells to maintain constant surveillance of the brain parenchyma for tissue changes, serving as a biosensor and a bioeffector [3]. When microglia sense a change in their surrounding environment, they can rapidly change their morphology by increasing soma size and retracting their normal elongated fine processes to shorter, coarser cytoplasmic processes displaying a bushy appearance. This was thought to continue to progress to a fully amoeboid morphology depending on the nature and severity of the signal and the biological process required [4,5]. This initial concept of a stereotypic response with limited response variability has been replaced by data showing that microglia respond with a variety of different morphological changes by integrating multifarious inputs [2,4,5,6]. Microglia actions may or may not require major morphological transitions, but rather more subtle changes in microglia morphology may reflect significant functional alterations. It is likely that the variety in morphologies reflects different functional changes of the cells or a difference in the nature of the initiation signal. With removal of the stimulating factor, microglia can then downregulate and return to the more ramified surveillant phenotype. The question remains however, as to whether this downregulated cell is representative of the “normal” cells, or whether the cell has been altered in some manner that may manifest in subsequent activation scenarios.

## 3. Cell Polarization

In general, macrophage responses are characterized based on the activating stimulus and the resulting production of factors [7]. A conceptual framework put forth for such activation identifies two polar extremes of signals: classical (M1) or alternative (M2) [8]. While this dichotomy has been found to be limited, as a working hypothesis this concept has been applied in an attempt to characterize microglial responses [9]. Lipopolysaccharide (LPS) and interferon gamma (IFNγ) are classic inducers of M1 activation [10]. LPS is recognized by toll-like receptor (TLR) 4, upon which activation induces myeloid differentiation primary response gene 88 (MyD88) and MaL/Tirap (Toll-interleukin (IL) 1 receptor domain containing adaptor protein)-dependent pathways for the expression of pro-inflammatory cytokines. IFNγ signals through IFNγ receptors (IFNγR1 and IFNγR2) for gene expression of cytokine receptors (IL-15R, IL-2RA, and IL-6R) and cell activation factors (CD36, CD38, CD69, and CD97). There are similarities between the gene expression profiles after IFNγ and LPS stimulation, yet they are not considered homologous [11]. Gene expression profiles following stimulation with a combination of the cytokines or various other stimuli are different from LPS or IFNγ profiles alone, generating a vast spectrum rather than a polarized dichotomy of responses [10]. Thus, the final production of factors depends highly on the activating stimulus and the surrounding environment.

In the early 1990s, the concept of alternative activation was developed largely based on work showing a role for IL-4 in the induction of an alternative (M2) activation state in peripheral macrophages [12]. Under this state, expression of anti-inflammatory cytokines (*IL-4, IL-10, IL-13*)—transforming growth factor beta (*Tgfβ*), arginase-1 (*Arg-1*), *CD206*, and Chitinase-3-like-3 (*Ym-1* in rodents)—was induced [6,13]. Upon further study, subclasses of M2 activation have been identified as dependent upon the inducting stimuli. The M2a activation state is induced by parasitic products or associated signals (IL-4 and IL-13), providing a longer-term function for resolution and repair [14,15,16,17]. In this case, signaling occurs through IL-4 receptor alpha (IL-4Rα), leading to inhibition of NF-κB signaling induced by M1 activation. M2b polarization is observed with triggering of Fc gamma receptors, TLRs, and immune complexes [18]. M2c polarization occurs in response to specific anti-inflammatory factors such as IL-10, TGFβ, and glucocorticoids [14,18,19]. In addition, cells do not appear to be committed to one phenotype, but rather can shift from M2b phenotype to a mixture of M1 and M2a/b [20]. M2 polarization of microglia is similar to peripheral macrophages [21,22,23,24], displaying different mRNA profiles for IL-4 and IL-10 stimulation, including *Arg1*; mannose receptor 1 (*Mrc1*); Y*m-1*, found in inflammatory zone 1 (*Fizz1*); and peroxisome proliferator-activated receptor (*Ppar*) [25]. The distinct polarization specificity of such markers remains in question. For example, *Arg1* is also induced in M1 macrophages [26]. In addition, induction of *Arg1* could be related to neurotoxicity based on arginine deprivation [27]. While these associations have been demonstrated in vitro, a number of the M2 related products can be induced in vivo by sterile wounds in the absence of IL-4 or IL-13 [28], suggesting an alternative stimulus. In vivo, a concurrent expression of M1 and M2 factors has been identified [27]. However, it is not yet clear whether the mixed expression pattern can occur within individual cells, or rather if the region of interest is comprised of cells showing different profiles.

From work assessing transcriptional regulation during human macrophage activation, an extended version of the M1 versus M2 polarization model was proposed that contained nine distinct activation programs [29]. As expected, a distinct axis of response was demonstrated upon stimulation with IFNγ or IL-4, or with stimulation with LPS and IL-13 polarization. A spectrum of signatures was identified with stimuli not directly linked to M1 or M2 polarization, such as free fatty acids, high-density lipoprotein, or combinations associated with chronic inflammation.

## 4. Microglia Signaling Processes 

The various functions of microglia are primarily accomplished through processes that involve cell signaling, phagocytic activities, cell-cell contact relationships, and secretion of various factors; while there are suggestions of unique differences, the general cellular processes are similar across the various regions of the CNS. As part of the innate immune system of the brain, microglia express a limited number of receptors or sensors on which multiple signals converge and upon which many effects manifest. Exogenous and endogenous signals with the capacity to mediate innate immune cell responses are comprised of the pathogen-associated “non-self” signals [30] and the “altered-self” danger associated signals [31]. These disparate signals may result in cell responses through common modes of action with specific receptor activation. Several highly enriched genes classified as “sensome” genes have been identified in microglia, which allows the cells to sense and interact with their local environment [32]. These sensome genes include those for putative purinergic receptors, P2ry12 and P2ry13; transmembrane protein 119 (Tmem119); G-protein coupled receptor 34 (Gpr34); the 1-type lectin receptor, sialic acid-binding immunogloubulin-type lectin H (Siglec-h); triggering receptors expressed on myeloid cells 2 (TREM2); and the fractalkine receptor, Cx3cr1. In comparison, peripheral macrophages have been shown to express a significantly lower number of these “sensome” genes, suggesting a less complex response network outside of the CNS [32]. 

### 4.1. Neuronal-Microglia Axis 

Microglia participate in a dynamic bi-directional communication with other neural cells. The predominant data available consider the communication between microglia and neurons. However, communications with astrocytes, oligodendrocytes, and pericytes are also likely contributors. The communication with neurons is considered to modulate and regulate microglia phenotypes via factors such a chemokines, neurotransmitters, and purinergic signaling. Microglia impact neuronal communication by actively monitoring neuronal function and sensing synaptic responses [33]. Even normal neuronal processes can have an effect on microglia. For example, increased neuronal activity can result in an increase in motility and physical contacts of microglia processes with neuronal elements [34]. This tightly regulated process ensures a balance of a surveillance state while maintaining the ability to react rapidly to a challenge. 

Neuronal immunomodulators such as CD200 are thought to maintain microglia in a homeostatic/surveillance state via CD200 receptor (CD200R) activation; however, this may also include active processes for regulation with activation of anti-inflammatory signaling [35]. A dysregulation in this ligand and receptor signaling has been reported across various neurodegenerative diseases [36]. CD200 is a type I membrane glycoprotein present on neurons and interacts with CD200R, a myeloid cell receptor on microglia [37]. A primary outcome of CD200/CD200R interaction is the control of many aspects of inflammation and immune responses [38,39]. The initial studies establishing the regulatory significance of this ligand/receptor interaction showed elevated neuroinflammation and immunoreactive microglia in CD200 knockout mice upon insult [37,38]. Work in mice suggested a role for this interaction to downregulate or silence immune functions of microglia under physiological conditions. These studies demonstrated a role for this interaction in regulating microglia activation and inflammatory responses, with CD200-deficient mice showing signs of an exacerbated response with injury or experimental autoimmune encephalomyelitis [23,37]. Disruption of CD200:CD200R signaling potentiated the pro-inflammatory response of microglia to immune stimuli [40,41] and exacerbated disease severity and progression in neuroinflammatory disease models [37,38,42], aging [43], neuropathic pain [44], and Alzheimer’s disease [45]. 

CX3CL1-CX3CR1 is a critical signaling pathway for microglia-neuron cross-talk [46,47]. CX3CL1 (also known as fractalkine) is a transmembrane protein that is a member of the chemokine CX3C class. It occurs in two isoforms: soluble and membrane-bound. In the CNS, CX3CL1 is primarily expressed in neurons within the amygdala, cerebral cortex, globus pallidus, hippocampus, striatum, thalamus, and olfactory bulb, with limited expression in the cerebellum [48]. CX3CR1 is the receptor for CX3CL1. It is expressed on the surface of monocytes/macrophages, neutrophils, T lymphocytes, natural killer cells, mast cells, thrombocytes, dendritic cells, and microglia [49,50,51,52]. Expression has been reported on microglia from early development throughout the lifespan [53], with an involvement in the induction of chemotaxis and cell adhesion. In addition, CX3CL1 is a factor secreted by medial ganglionic eminence interneurons, and is necessary to promote cortical oligodendrogenesis [54]. Experimental genetic ablation of CX3CR1 resulted in increased microglia activation [55], suggesting that the CX3CL1/CX3CR1 signaling axis keeps microglia in a homeostatic phenotype under physiological conditions. The CX3CL1-CX3CR1 axis in aged microglia contributes to processes associated with CNS aging and age-associated neurodegenerative diseases [56]. 

The neuron-derived factor, CD22, and the microglia transmembrane protein-tyrosine phosphatase, CD45, are an example of ligand-receptor interaction that serves to suppress microglia responses. Binding of this receptor inhibits the production of pro-inflammatory molecules in response to lipopolysaccharide (LPS) [57]. An additional ligand/receptor interaction that can suppress pro-inflammatory cytokine production is the signal regulatory protein (SIRPα), which is expressed on myeloid cells, neurons and astrocytes, and CD47, which is expressed on microglia and neurons [58].

### 4.2. Pattern Recognition Receptors

The tight regulation of neuroimmune cells, primarily microglia, is accomplished by a number of signaling pathways, as reviewed by ElAli and Rivest [59]. As the major line of defense in the brain, microglia express several receptors involved in the control of innate immune functions. Similar to the process in the periphery, basic host defense mechanisms in the CNS begin with the recognition of warning signals generated by PAMPs such as bacterial, viral, and protozoal products (protein lipid, nucleic acid, and carbohydrate). Microglia detect ligands for CD40, CD91, and the intracellular NOD-like receptors (NLRs) that initiate the signaling process by binding to PAMPs. Depending on the stimulus, inflammatory responses can be initiated by pattern recognition receptors (PRRs). Relevant PRRs include major receptor families such as TLRs, the receptor for advanced glycation end products (RAGE), C-type lectin receptors (CLRS), RIG-like receptors, AIM2-like receptors, and scavenger receptors to detect the aberrant expression of phosphatidylserine on the extracellular surface of dying cells. The engagement of these receptors and the subsequent signaling pathway activation serve to tailor the innate response to the activation stimulus. 

In a sterile inflammatory response, immune cells are activated in the absence of microbial compounds. Under these conditions, immune cells detect tissue damage and induce sterile inflammation upon the binding of endogenous ligands or DAMPs released by stressed or injured cells [60]. These include nucleic acids, lipids, and proteins that normally are not present to immune cells until released or unmasked with cell injury or death. Such factors include RNA released by necrotic cells, released mitochondrial DNA, the nuclear chromatin protein high mobility group box 1 (HMGB1), heat shock proteins, α-synuclein, silica particles, and amyloid β peptides derived from the amyloid precursor protein. Intracellular DAMPs include such factors as HMGB1 and peroxiredoxin (Prx) family proteins. Factors in the extracellular matrix can also function as DAMPs. As examples, fibrinogen and proteoglycans (hyaluronan, biglycan, and versican) can activate the TLRs and initiate an inflammatory response [61,62,63]. 

#### 4.2.1. Toll-Like Receptors

The toll-like receptor (TLR) family is a major class of PPRs located on the plasma membrane or in endosomal compartments [64]. Members of this family contribute to the initiation and tailoring of innate and adaptive immune responses. The different TLRs are activated by distinct PAMP epitopes to engage specific downstream intracellular signaling cascades [65]. These specificities range across bacterial DNA CpG motifs (TLR9), gram-negative lipopolysaccharide (TLR4), gram-positive peptidoglycan (TLR2), viral double-stranded RNA (TLR3), and fungal zymosan (TLR2). 

For the classical TLR signaling events, ligand binding induces a conformational change in the receptor, allowing for an association with the MyD88 intracellular adapter protein. Once bound to the receptor, IL-receptor associated kinases (IRAK) 1 and 4 then associate with MyD88 through the death domain. This leads to phosphorylation of IRAKs, which then oligomerize with the tumor necrosis factor (TNF) receptor associated with factor-6 (TRAF-6) for activation and polyubiquitination of the oligomer. Transforming growth factor β-activated kinase (TAK1) is then employed to activate IκB kinase (Iκκ) to tag IκB for degradation, allowing for nuclear translocation of NF-κB. This leads to the production of proinflammatory cytokines and chemokines. While NF-κB activation is well defined, functions for MyD88 include activation of other transcription factors, such as IRF1, IRF5, and IRF7 [50]. This is the primary pathway for TLR2 signaling, while TLR3 relies on a toll/IL-1 receptor (TIR)-domain-containing adapter-inducing IFN-β (TRIF)-dependent signaling pathway [66]. TLR4 utilizes both the MyD88-dependent and the TRIF-dependent signaling pathways, and is responsible for recognizing the gram-negative cell wall component, lipopolysaccharide. This potent stimulus for microglial activation is typified by the robust production of numerous proinflammatory factors. In addition, CD14 interacts with TLR4 to induce maximal responses to lipopolysaccharide.

In the nervous system, a wide spectrum of TLRs is represented on microglia while a more limited number is expressed on astrocytes. Excellent overviews of TLRs in the CNS are available [67,68,69,70,71]. In human microglia isolated from autopsy material, expression of TLR2 and TLR3 mRNA and protein has been detected with only mRNA levels for TLR4 and not protein [72]. This pattern was evident under naïve conditions and upon stimulation by receptor specific factors. In this study, human microglia expressed *IFN*-β and secreted significant levels of CXCL-10 upon TLR3 and TLR4 activation. Neither were observed with TLR2 activation; however, a release of TNFα and IL-6 was observed. In addition, signaling via each of these TLRs can upregulate TLR2 expression, and signaling through TLR3 and TLR4 upregulated TLR3 expression, suggesting an activation-induced positive feedback loop. In contrast, stimulation of TLR2, TLR3, or TLR4 resulted in a downregulation of TLR4. TLRs have also been identified on astrocytes [73]. Astrocytes isolated from human autopsy tissue were found to express TLR3 mRNA and protein. TLR2 and TLR4 were observed at the mRNA level but not as protein [72]. In response to TLR3 signaling, astrocytes released IL-6, CXCL-10, and increased mRNA levels for IFN-β. The specificity of TLR3 in astrocytes was also reported by Farina et al. [74] showing elevations with IFN-γ, IL-1β, and IFN-β. Of additional interest is the earlier report that human neurons express only mRNA for TLR3 [75] and TLR8 [76]. These findings suggested that the immune responses triggered in and by microglia and astrocytes were distinct and tailored to the environmental signal. 

#### 4.2.2. Receptor for Advanced Glycation End Products (RAGE)

RAGE is a multiligand receptor belonging to the immunoglobulin superfamily that is involved in numerous cell processes, including neuroinflammation, apoptosis, proliferation, and autophagy. It is expressed in multiple neural cell types, including neurons, microglia, astrocytes, and vascular endothelial cells. Activation occurs with production of advanced glycation end products in pro-oxidant and inflammatory environments. It recognizes other ligands, including serum amyloid A, S100 protein, and HMGB1. Similar to other immune receptors, binding of RAGE induces a series of signal transduction cascades, leading to the activation of NF-kB and pro-inflammatory cytokine release. RAGE contributes to the clearance of amyloid β and is involved in apolipoprotein E mediated cellular processing and signaling. Recent work suggests that RAGE is associated with the response to stress and depressive-like behaviors [77]. 

#### 4.2.3. NOD-Like Receptors (NLR)

The NLR (nucleotide binding domain, leucine rich repeats-containing) family of genes and proteins is heavily implicated in regulation of immunity. NOD-like receptors (NLRs) are soluble cytoplasmic PRRs that act as sensors of cellular damage and effectors of inflammation. Their function is dependent on the assembly of large (~700 kDa) complexes termed “inflammasomes”. The largest NLR subfamily, and the one most pertinent for neuroinflammation, is designated the NACHT domain-, LRR domain-, and pyrin domain-containing protein (NALP) family or NLRP3 [78]. While multiple inflammasome protein complexes have been demonstrated, NLRP3 remains somewhat unique in that while it responds to a broad spectrum of exogenous and endogenous activators, it generally requires transcriptional cell priming by an activating ligand prior to activation [79,80]. This step involves an NF-κB-dependent upregulation of cellular NLRP3, pro-IL-1β transcription, and *de novo* protein synthesis upon recognition of pro-inflammatory stimuli and TLR activation. Once primed, NLRP3 activation can be induced by a variety of extracellular, sterile, non-pathogenic triggers, most of which work through activation of purinergic receptors or ionic membrane pore alterations. These sterile activators, or “triggers”, include cholesterol crystals [81] and uric acid crystals [82,83], aggregated proteins and lipids [84,85], silica and asbestos [86], aluminum salt adjuvant [87], polystyrene nanoparticles [88], and tri-organotin compounds [89]. In addition to these extracellular signaling factors, the release of mitochondrial DNA may elicit an inflammasome response [90]. Following stimulation, NLRP3, the common adaptor apoptosis-associated speck-like protein containing a CARD (ASC), and an effector, caspase 1, combine to form the inflammasome complex. Within this complex, pro-caspase-1 is activated, which in turn cleaves and activates the pyrogenic cytokines, IL-1β and IL-18, by cleavage from the pro-form to the mature form of the protein [91,92]. This then leads to either a pyroptotic cell death and/or the excretion of exosomes [93,94], or the oligomeric NLRP3 inflammasome particles [95,96]. These factors can act on adjacent cells to activate the NF-κB signal pathway, alter lysosome integrity, and enhance or prolong the immune response [97]. It is thought that cooperative signaling via TLRs and NLRs resulted in secretion of the IL-1 family cytokines. While inflammasome activation is an efficient producer of mature IL-1β, inflammasome independent mechanisms for the production of mature IL-1β include cathepsin B or caspase 11 dependent pathways, bacterial pore-forming toxins, and extracellular ATP [98,99]. Thus, an upregulation of mature IL-1β does not necessarily indicate an inflammasome mechanism.

The mechanism by which NLRP3 is activated by the secondary “trigger” is not fully understood. However, there is mounting evidence that mitochondrial damage plays a central role. As an example, the complex 1 inhibitor, rotenone, can serve as a secondary trigger for NLRP3 inflammasome activation [100]. Conditions that have been found to facilitate NLRP3 activation include increased ROS production, calcium influx, potassium efflux, reduction in NAD+, externalization of cardiolipin from mitochondria, and presence of mitochondrial DNA in the cytoplasm, all of which are the result of mitochondrial dysfunction and damage [101,102]. Thus, it is not surprising that NLRP3 inflammasome activation has been implicated in various neurological diseases [103] and is a contributing factor to inflammaging [104]. One regulator process identified is the Parkinson’s disease-associated mitochondrial serine protease, HtrA2, which has been shown to restrict activation of ASC-dependent NLRP3 and AIM2 inflammasomes, in a protease activity-dependent manner [105].

#### 4.2.4. Triggering Receptor Expressed on Myeloid Cells 2 (TREM2)

TREM2 is a PPR specific to polyanionic that is located on the membrane surface of osteoclasts and microglia [106]. Activation of TREM-2 receptors contributes to an upregulation of chemokine synthesis and phagocytosis of apoptotic cell debris [107]. This is accomplished upon binding with the DNAX-activation protein 12 (DAP12), an ITAM-containing adaptor protein, triggering reorganization of F-actin, and phosphorylation of ERK/MAPK [107,108]. Signals are transmitted via rapid phosphorylation of the immunoreceptor tyrosine-based activating motif (ITAM) of DAP12, mediated by Src protein tyrosine kinases, followed by the binding of phosphorylated ITAM to Src homology 2 (SH2) domains of spleen tyrosine kinase (Syk), resulting in autophosphorylation of the activation loop of Syk [109]. The expression of TREM2 is downregulated by LPS and interferon (IFN)-γ [110]. TREM2 is a key negative regulator of autoimmunity and plays a role in the inhibition of IL-6 and TNF production by macrophages [106]. In addition, it is responsible for DAP12-induced inhibition of inflammatory responses driven by TLR agonists in mouse and human macrophages. There is also some evidence that TREM2 activates signal transduction pathways that promote microglia chemotaxis, phagocytosis, survival, and proliferation. TREM2 receptor expression can contribute to apoptotic neurons in the absence of inflammation [107], detect damage related to lipid patterns, and contribute to the response of microglia to amyloid-β accumulation [111]. The TREM2-DAP12 axis plays a role in the function of aged microglia, and thus is associated with changes observed in physiological CNS aging or neurodegenerative disease [56,112,113]. A direct relevance to these signaling and functional alterations in microglia is demonstrated in Nasu–Hakola disease [114,115]. In this loss-of-function disease, genetic mutations of TREM-2 and DAP12 occur that result in aberrant TREM-2/DAP12 signaling pathways. Neuronal expression of phosphorylated Syk is enhanced in the cerebral cortex and the hippocampus of Nasu-Hakola brains [109], and microglia show a diminished capacity for phagocytosis. This disease is associated with progressive presenile dementia and sclerosis in the front-temporal lobe and the basal ganglia [114,115]. Interestingly, TREM2 expression is elevated in the cognitively normal brain with aging [116], and is considered to be a possible compensatory mechanism against age related deterioration of microglia function.

## 5. Microglia Phagocytosis

Phagocytosis is a critical function provided by microglia; however, it is also the function that raises the most questions as to whether the actions are beneficial or detrimental. Through phagocytosis, microglia significantly contribute to the health of the nervous system by clearing excess cellular debris, aberrant proteins, and invading bacterial or viral fragments. The phagocytic actions of microglia facilitate structural and functional development of the nervous system, as well as recovery of impaired tissue to maintain a homeostatic balance. It is difficult to distinguish signals for pathological phagocytosis from normal phagocytosis, or even diminished phagocytosis. It is also the divisive property of microglia that can lead to an association with beneficial or detrimental effects. It is thought to be provoked by “eat-me” signals or “don’t eat me” signals provided by the viable cell and regulated by the identification of “self” or “non-self”. While it has been well established that microglia serve in this function to clear material from the brain, there are data suggesting that cell death of viable cells can be initiated by microglia phagocytosis. This has been termed “phagoptosis” [117]. Negative effects of microglial phagocytosis can be driven by unwarranted or misdirected phagocytosis that can either be excessive phagocytosis or reduced phagocytosis [118]. Excessive phagocytosis of synapses, neuronal cell bodies, or myelin sheath can be detrimental to tissue and functional recovery. In contrast, reduced microglial phagocytosis, such as reduced phagocytosis of synapses or clearance of unhealthy neural cells, may lead to pathological neural connectivity. In addition, a diminished ability to clear aberrant proteins or cellular debris from the neural environment would also be damaging to the structure and functioning of the system. 

There are a number of steps involved in successful phagocytosis. The initial one relies on chemotaxic signals to allow the microglia to migrate to the target site. This is then followed by a signal on the target cell that identifies it for phagocytosis, often involving the Ras homologous (RHO) protein. Once the microglia anchors to the target, the cell must internalize and engulf the target, which is a process that often involves cell cytoarchitectural proteins. Once internalized, phagosomes that contain the material become acidic and fuse with lysosomes for enzyme degradation. Phosphatidylserine (PtdSer) is the most common membrane-anchored “eat me” signal in the nervous system. Exposure of PtdSer marks the neuron for selective engulfment [119]. While expression of PtdSer on the surface can be reversed, the surface expression still represents a stressed neuron, and thus, microglia would be appropriately responding to the signal. Polysialylated proteins on neurons also inhibit phagocytosis by binding to receptors on microglia, and by activating sialic acid-binding immunoglobulin-like lectins (SIGLECs), such as SIGLEC-11 (in humans) and SIGLEC-E (in mice) [119]. 

CD36 is a raft-resident, cell surface glycoprotein with expression limited to specific cellular subtypes including monocytes and macrophages [120]. It functions as a scavenger receptor for which extensive glycosylation of CD36 is required for intracellular trafficking onto the cell membrane [121]. Its involvement in phagocytosis was first reported by Ren et al. [122] in the clearance of apoptotic cells. It is also involved in phagocytic processes for the clearance of necrotic cells [123] and myelin [124]. Membrane raft aggregation in the phagocytic cup and recruitment of CD36 has been reported to be necessary for microglial phagocytosis of Aβ_42_ [125]. Recent work demonstrated CD36-TLR4-TLR6 activation as a common mechanism by which atherogenic lipids and amyloid-beta stimulate sterile inflammation, suggesting a new model of TLR heterodimerization triggered by co-receptor signaling events [126].

The integrin-associated protein, CD47, is a receptor for thrombospondin family members and a ligand for the transmembrane signal-regulatory protein (SIRP) alpha. It is expressed on myelin, myeloid cells, red blood cells, platelets, neurons, fibroblasts, and endothelial cells [127]. The role of CD47 in phagocytosis and immune recognition was discovered in red blood cells showing that the CD47-SIRPα axis plays a role in self-identifying and protecting red blood cells from phagocytosis [128]. In addition, it is upregulated on circulating hematopoietic stem cells to protect against phagocytosis [129]. Oncogenic activation of CD47-SIRPα signaling appears to enable cancer cells to evade immune detection and clearance [130]. Studies have suggested that CD47 may localize to synapses [131], and that CD47-SIRPα protects developing synapses from aberrant removal [132]. In cultured microglia, CD47 has been shown to re-enter lipid membrane rafts in microglia during phagocytic inhibition [125]. Polysialylated proteins on neurons inhibit phagocytosis by binding to receptors on microglia and by activating sialic acid-binding immunoglobulin-like lectins (SIGLECs), such as SIGLEC-11 [in humans] and SIGLEC-E (in mice) [120]. 

Calreticulin is localized in the endoplasmic reticulum; however, upon surface exposure, binding to the low-density lipoprotein receptor-related protein located in microglia induces phagocytosis [133]. Such soluble bridging molecules require at least two binding domains to serve as a link between the membrane-anchored signal and phagocytic receptors. During inflammation, milk fat globule epidermal growth factor 8 (MFG-E8) released from microglia or astrocytes can bind to exposed PtdSer and the vitronectin receptor. Mer tyrosine kinase (MerTK) is upregulated by inflammatory cytokines and acts as a microglial phagocytic receptor to mediate phagocytosis of apoptotic cells, stressed neurons, and synapses. MerTK interacts with PtdSer through two soluble bridging molecules, Gas6 and Protein S. For these molecules, their N-terminal 11 γ-carboxyglutamic acid residues can bind to PtdSer [134]. Annexin 1 (ANXA1) is sparingly expressed in microglia of normally aged human brains, with a higher level of expression in Alzheimer’s disease. When released from microglia, ANXA1 can bind to neuronal PtdSer and activate microglial formyl peptide receptor 2 [135]. Using *in vitro* systems, two distinct roles for ANXA1 have been identified. The first was for controlling the non-inflammatory phagocytosis of apoptotic neurons and the second for promoting resolution of inflammatory microglial activation [135].

Adenosine triphosphate (ATP) and other purine and pyrimidine nucleotides are released upon cell injury. As ligands, they trigger membrane-bound purinergic receptors to regulate several physiological processes [136,137,138,139]. These processes include immune cell recruitment, inflammation, and neurotransmission [140,141]. A dysregulation of the purinergic pathways has been implicated in neuroinflammatory responses and neurodegeneration [142]. The release of ATP induces chemotactic and chemokinetic activities of microglia through stimulation of P2Y4R. Both ATP-gated P2X4R and UDP-activated P2Y6R are upregulated in activated microglia following neuronal injury. P2Y6R decreases P2X4R-mediated calcium entry and inhibits the dilation of P2X4R channels into a large-conductance pore [143]. Upon P2Y6R receptor activation, phospholipase C is activated and inositol 1,4,5-triphosphage is synthesized, leading to the release of Ca2+. This then stimulates phagocytosis. UDP-induced P2Y6R stimulation can prevent the ATP-dependent migration of microglia, most likely by switching from its migratory phenotype to a phagocytic one [144]. ATP acting on P2X7Rs decreases phagocytic capability and promotes activation and proliferation of microglia [145,146]. It has been reported that the ATP-gated ionotropic P2X7 receptor functions as a small cation channel and can trigger permeabilization of the plasmalemmal membrane [147]. 

The primary focus has so far been on phagocytosis of injured or dead neurons, or aberrant proteins from the brain, but phagocytosis of synapses is a more targeted process of interest. Stripping of synapses is a critical process during development, in learning and memory, and in neurodegeneration. The question is whether this is a process initiated by the microglia, as implied by the term “stripping”, or rather a facilitatory action by microglia as signaled by the neuron [148]. One method by which microglia facilitate the removal and clearance of synaptic debris is through the complement system. While much of the work demonstrating a functional association between complement expression and microglia activation at synapses has focused on the developing nervous system, reports of an involvement of complement activation in injury models have established a basis for translating these interactions to adults [149,150,151]. How these processes relate to phenotypic changes in resident microglia still remains in question; however, C1q has been considered a likely candidate to drive microglial activation. Components of the complement system facilitate immune responses [152], and have been considered as likely candidates for mediating “on” signals for microglial activation, potentially by opsonizing or tagging the target structure for elimination [153]. A number of downstream effector functions are initiated with complement activation, including surface deposition of opsonin C3b on target cells, recruitment of phagocytic cells, and pathogen lysis [154]. An elevated expression during injury has been proposed as a mechanism to target microglia to degenerating cellular processes, and an association between microglia and active synaptic loss or remodeling has been reported in various brain regions [155,156,157]. A key component to initiate this cascade is the complement protein, C1q, which functions as a recognition component of the macromolecular complex, C1. Microglia and infiltrating blood-borne monocyte/macrophage cells serve as primary sites of C1q synthesis in the CNS [158,159,160]. A spatial and temporal synergy between actions of microglia and the distribution of complement factors has been suggested to represent signaling interactions for targeting microglia actions [161], not only for apoptotic neurons but also for rapid elimination of synapses [162,163,164]. C1q can detect PtdSer exposure and subsequently bind to PtdSer, resulting in the removal of synapses. C1q, both alone and with C3, can facilitate microglial clearance of misfolded proteins, apoptotic neurons, and neuronal blebs on damaged cells [161,164,165].

## 6. Microglia in Neurodegeneration and Brain Injury

Neuroinflammation, mediated largely by microglia, has been implicated in several different neurological disorders from acute injuries, such as stroke or traumatic brain injury, to chronic neurodegenerative conditions, such as Parkinson’s disease (PD) or Alzheimer’s disease (AD). In each of these conditions, there are varying degrees of involvement of infiltrating blood-borne immune cells that complicate the ability to identify specific effects of resident microglia from those of peripheral macrophages. In progressive neurodegenerative diseases, the early stage of the disease likely involves recruitment of resident microglia, while with increasing severity, the BBB may become compromised and allow infiltration or targeted recruitment of peripheral macrophages. Determining the cellular source of the inflammatory response is often difficult given the inability to easily distinguish between resident microglia and infiltrating macrophages.

Genome-wide association studies of patients with PD or AD have implicated mutations in genes highly expressed on microglia [166,167,168]. The ubiquitous nature of microglia and neuroinflammation across a broad spectrum of neurodegenerative diseases or brain injury makes it difficult to discriminate between those responses that lead to a detrimental outcome versus those that simply represent microglia performing their normal functions to return the system to homeostasis. Thus, it remains unclear if microglia function in neurodegenerative diseases is beneficial but insufficient or if the beneficial aspect diminishes with the progression of the disease and the severity of the neuronal damage.

Efforts to identify these differences identified a unique population of microglia within models of human neurodegenerative disease that have been named Disease Associated Microglia (DAMs) [169,170,171]. This population represents microglia in the initial stages of DAM activation that inhibit microglia checkpoints and are TREM2 independent, while progression to a full activation of the DAM phenotype is TREM2-dependent and includes phagocytic and lipid metabolism activity. It has been speculated that this distinct microglia phenotype has the potential to restrict neurodegeneration. This protective response is proposed to be due to a dedicated sensory mechanism, including TREM2, to detect damage within the CNS as identified from neurodegeneration-associated molecular patterns (NAMPs) [170].

### 6.1. Parkinson’s Disease

Parkinson’s disease (PD) is a member of the progressive adult-onset neurodegenerative diseases classified as α-synucleinopathies. These include PD, dementia with Lewy bodies (DLB), and multiple system atrophy. The main pathological hallmark of these diseases is the occurrence of hyperphosphorylated, misfolded and fibrillized α-synuclein positive inclusions known as Lewy bodies that can be observed throughout the CNS. In PD and DLB, neurons are the main cell type displaying cytoplasmic α-synuclein positive aggregations, the major component of Lewy bodies and Lewy neurites. Alpha-synuclein deposits can also be observed in astrocytes and oligodendrocytes [171]. Pathological hallmarks of PD include the loss of dopaminergic neurons in the substantia nigra pars compacta and the presence of Lewy bodies in the surviving dopaminergic neurons.

In the brain, α-synuclein is predominantly located in presynaptic terminals of neurons in the hippocampus, striatum, thalamus, cerebellum, and neocortex [172]. α-synuclein can exist in several forms, including soluble unfolded monomeric and polymeric forms, as well as β-sheet-containing fibrils [173]. The ability to self-aggregate is considered a pathological role of α-synuclein and, as a secondary process, the fibrillar form can serve as seeding material for α-synuclein aggregation [174]. In addition, α-synuclein overexpression impairs lysosomal proteolytic clearance of damaged cellular material, including damaged mitochondria, resulting in an accumulation of abnormal proteins or damaged organelles. Both macroautophagy and chaperone-mediated autophagy appear to be compromised in PD, suggesting that a reduced clearance of α-synuclein contributes to the generation of α-synuclein inclusions [175]. The need for successful phagocytic clearance and the response of the nervous system to accomplish this was demonstrated in the upregulation of TLR4 in human multiple system atrophy patients, suggesting an attempt to increase phagocytotic activity [176]. The spreading of α-synuclein with Lewy body-like pathology to anatomically interconnected regions may be mediated by several mechanisms of cellular release and uptake, including exocytosis, exosomes, tunnelling nanotubes, glymphatic flow and endocytosis [177]. It has been demonstrated that α-synuclein overexpressing neuronal cells can release exosomes capable of transferring α-synuclein protein to other normal neuronal cells [178]. Within these new neurons, they can form aggregates and induce cell death [179,180]. It has been demonstrated that exosome-associated α-synuclein oligomers are more likely to be taken up by recipient cells and are more toxic than free α-synuclein oligomers [181]. This release of exosomes may serve to trigger an early microglia response; in addition, recent work has suggested a role for microglia in spreading and transfer of misfolded α-synuclein via exosomes [182,183]. 

The association between inflammatory signs, cellular response, and neuronal degeneration in PD has received attention over the last decade, demonstrating a complex interaction between glia and alpha-synuclein and neuronal derived DAMPS [184,185]. The hypothesis that microglial cells can be activated by extracellular α-synuclein or astroglia and that this can occur prior to neuronal loss in the substantia nigra pars compacta but concurrent with neuronal dysfunction and loss of dopaminergic terminals, has been proposed [186,187]. Su et al. [186] reported activation of primary rodent microglia by exogenously mutated α-synuclein and Halliday and Stevens [187] proposed that astrocytic α-synuclein initiated a microglia response. 

Attempts to image neuroinflammation and microglia activation in life has relied on binding ligands for translocator protein (TSPO), formerly known as the peripheral benzodiazepine receptor. This is an 18 kDa outer mitochondrial membrane protein that shows robust binding in the periphery and limited signal in the CNS parenchyma. In healthy young human subjects, constitutive TSPO protein expression can be observed throughout the entire brain. This constitutive binding has complicated identification of a valid non-binding reference region. Cautions have been raised with the general use of TSPO expression as a diagnostic biomarker and therapeutic target for a broad range of inflammatory, neurodegenerative, and psychiatric disorders, and with interpretation of the data available across multiple generations of probes [188,189,190].

In the early 2000s, Positron Emission Tomography (PET) imaging studies reported evidence of microglia activation in PD [191]. It was thought that the imaging data supported this temporal hypothesis; however, a signal was detected in all brain areas implicated in PD [192]. PET imaging with the peripheral benzodiazepine receptor binding ligand [11C]-[R] PK11195 indicated that, irrespective of the number of years with the disease, patients with idiopathic PD had a markedly elevated signal in the pons, basal ganglia, striatum, and frontal and temporal cortical regions as compared with age-matched healthy controls [171]. In comparison, Ouchi et al. [193] showed that, with disease progression, binding of [^11^C]CFT to the dopamine transporter decreased and the TSPO signal spread over the entire brain, eliminating indications of regional specificity. These findings were obtained with a first-generation probe [[11]C]-PK11195 for which subsequent studies using a second generation probe did not readily substantiate [194]. More recent work, using [^18^F]-DPA714 and adjusting for TSPO polymorphism, reported higher levels of binding in the midbrain, frontal cortex, and putamen of PD patients [195]. 

Much of the work regarding in life imaging or post-mortem analyses of microglia involvement in PD comes from the patient population making it somewhat difficult to distinguish initiating effect from a reactive process or a contributing process. In PD autopsies, a morphological response of microglia was observed in the substantia nigra that was not accompanied by a change in astrocyte morphology or glial changes in the putamen [196]. Work from Doorn et al. [197] showed microglia morphological heterogeneity in the hippocampus and substantia nigra in post-mortem analysis of PD brains. In addition, the expression of TLR2 on microglia was elevated in incidental Lewy body disease cases. Based on experimental studies, it was suggested that the presence of α-synuclein in the Lewy bodies triggered the microglia response via TLR2. Work from Reynolds et al. [198] reported NF-kB related inflammatory processes in the substantia nigra and basal ganglia of PD patients. Other inflammatory-related indicators have been reported in the post-mortem PD patient, including human leucocyte antigen type DR [HLA-DR+] [199] and CD68 [200], cyclooxygenase (COX), and inducible nitric oxide synthase (iNOS) [201]. Imamura et al. [199] reported an increase in the number of MHC class II-positive microglia in the SN and putamen with PD. This was also observed in the hippocampus, trans-entorhinal cortex, cingulate cortex, and temporal cortex, and persisted regardless of the presence of Lewy bodies. These researchers reported that the MHCII positive microglia co-expressed TNFα and IL-6, and were associated with α-synuclein positive Lewy neurites, tyrosine hydroxylase 16 positive dopaminergic, WH-3 positive serotonergic, microtubule associated protein-2 positive neurites, and SMI32 positive neurites. 

Thus, there are a number of signaling processes that can induce a response from microglia from the earliest stages of neuronal dysfunction to the final neuronal death and clearance stage. In addition to inflammatory factors and growth factors, some evidence suggests that exosomes released from microglia have an active role in α-synuclein transmission. Different cell types within the brain have been shown to release exosomes, including neurons, microglia, and astrocytes. Microglia have been observed to efficiently secrete exosomes as part of their antigen presentation and cargo release mechanisms. α-synuclein is found in the exosomes from the microglia BV-2 cells, which have been shown to cause apoptosis in neurons [202]. This is consistent with another study showing that misfolded tau protein, an important pathology in Alzheimer’s disease, can spread via microglial exosomes, whereas depletion of microglia and inhibition of their exosome synthesis halt tau propagation [203]. 

There is clear evidence of a role for microglia and neuroinflammation; however, remains to be determined whether this implicates these processes in the initial injury, the attempted anti-inflammatory and repair, or the excessive demand above the capacity of microglia to perform their function to return the brain region to a homeostatic state. What has become evident over the years is the possible contribution from the periphery. Elevated levels of cytokines have been demonstrated in the peripheral circulation of PD patients, including the pro-inflammatory cytokines, TNFα and IL-1β, and the anti-inflammatory cytokines, IL-10 and IL-4 [204,205,206]. It is likely that the death of dopaminergic neurons along with the local environment created by cytokine signaling may facilitate the recruitment of peripheral immune cells to impact survival of nigral DA neurons and progression of PD. Efforts have been conducted to try to identify an association between inflammation and PD.

### 6.2. Stroke

Cerebral ischemia is a multiphasic process that triggers acute inflammation that exacerbates the primary damage to the brain. Stroke represents many of the issues in all neurodegenerative or injury models with regards to the need to distinguish between resident glia responses and those from blood-borne cells that are allowed to infiltrate into the brain parenchyma. As such, it offers an acute injury directly associated with a localized insult to the vascular wall for examination of the distinction between resident microglia and infiltrating macrophages. The BBB excludes plasma proteins and many of the peripherally derived innate and adaptive immune cells, as well as their associated inflammatory molecules. With compromise to the functional permeability of the BBB, blood-borne monocytes can enter the brain parenchyma and assume a brain macrophage phenotype, making them difficult to distinguish from resident microglia [207,208,209]. These infiltrating cells are considered to be predominantly involved in severe inflammatory injuries, while resident microglia focus on tasks related to maintaining tissue homeostasis [210]. 

The regulation of inflammation after stroke is multifaceted and comprises vascular effects, distinct cellular responses, apoptosis and chemotaxis. There are many cell types that are affected, including neurons, astrocytes, microglia and endothelial cells, all responding to the resultant neuroinflammation in different ways [211]. Microglia activation has not been clearly demonstrated to either facilitate or hinder recovery from stroke. Data has shown that TNFα can actively protect neurons [212] and that mice deficient in TNFp55 receptors show increased stroke volume [213], while overexpression of TNF-α is detrimental to stroke outcome [214].

The contribution of inflammation to secondary brain damage is supported by several studies. Within the first few hours, a rapid activation of resident microglia and production of proinflammatory cytokines occurs, including TNFα and IL-1β [215]. This is followed by a progression of ischemic brain injury associated with an intense inflammatory response. Neutrophils and monocytes/macrophages infiltrate and accumulate in microvessels and ischemic cerebral parenchyma [216]. The studies of Yilmaz and Granger [217] suggested that lymphocytes can be recruited into the site within the first 24 h. Work by Gelderblom et al. [218] provided a detailed characterization in mice of the temporal dynamics of immune cell accumulation following transient middle cerebral artery occlusion. While this is a rodent response, the concepts of timing for evaluating resident microglia responses apply across species. With examination of animals at 12 h and days 1, 3, and 7, a widespread neutrophilic infiltration into the ischemic hemisphere was observed at 3 days. Infiltration was also seen in the contralateral hemisphere, but at a much lower level. Iba-1 immunopositive microglia/macrophages were observed as early as 12 h, increasing after 24 h, and were in close proximity to the infarct, while a large number of CD11c immunopositive dendritic cells were located in the ischemic hemisphere close to the penumbra. Only after 12 and 24 h were neutrophilic granulocytes observed. This was at a minimal level, and by day 3 these cells were extensive in the ischemic hemisphere, equating to the number of microglia/macrophages. Of interest to the focus of this review was the response in sham-operated animals. In this case, microglia increased on day 1; however, by day 3, blood-derived immune cells were the predominant cell type, suggesting that the anesthesia and surgical manipulation alone were sufficient to induce a level of change. These infiltrating cells included neutrophils, macrophages, dendritic cells, natural killer cells, and lymphocytes. While consistent with previous reports, the peak time for neutrophil influx occurred later than previously reported [216].

### 6.3. Aging as a Co-Factor for Neurodegeneration and Microglia 

The predominant risk factor for neurodegenerative disease is age, and thus, represents a significant co-morbidity. Thus, understanding age-related changes in brain immune cells and the impact of those changes is critical in addressing the contributory role of these cells and processes to any neurodegenerative disease process [219,220,221,222]. A common feature of aging is “inflammaging” [223]. This is characterized as a low-grade, clinically undetectable inflammation that occurs as the result of an unbalanced regulation of the immune system. It was initially considered as a disequilibrium between the innate and adaptive immune systems; however, this is not an equally distributed deficit, as it appears that the adaptive immune system is more vulnerable. In the nervous system, such primary challenges would be related to the imbalance between proteostasis and autophagy—the removal of the accumulated cellular garbage of the brain.

Microglia are long-lived cells, and as such, they represent immune cells that show some sort of memory, which for peripheral immune cells has been termed “trained innate immunity” [224,225,226,227]. Following a response to a pathogen-associated molecular pattern [PAMP] or to a danger-associated molecular pattern [DAMP], the cells return to a quiescent/surveillance state; however, they can retain some epigenetic/molecular changes that constitute their “training” [226]. Often this is accompanied by a modified pattern of resting and activation-induced secretion of proinflammatory and anti-inflammatory cytokines [225]. While this has been demonstrated in systemic immune cells, exactly how this translates to cells within the CNS is not clear but would be expected to follow similar processes. It is likely that the elevation of a pro-inflammatory phenotype of microglia that occurs in the CNS with aging represents this type of training.

Each of the major effector functions of microglia show a deterioration with aging. Age-related deterioration can be represented by loss-of-function, hyper-reactivity, or dysfunction, each of which would have an impact on the timing of onset and the progression of neurodegenerative diseases. The aging of microglia may be the result of a cumulative activation over time due to local events or systemic infections and inflammation [228]. The effects that occur with aging may be exacerbated by a neurodegenerative disease phenotype, genetic variations, or other predisposing factors. Early work examining the morphological characteristics of microglia in the aged brain suggested a shift in morphology with a greater occurrence of dystrophic microglia processes [229,230]. The shift in morphology was accompanied by a decrease in microglial process motility [231,232]. In general, the morphological aspects of microglia in aging coincide with an impairment of signaling to maintain microglia in a surveillance state, an impairment of neuroprotective functions, and an enhancement of neurotoxic responses [233,234]. One critical feature of senescent cells, apart from cessation of divisions, short telomeres, morphological changes, and activity of the SA-βGal enzyme, is the senescence-associated secretory phenotype. This is the ability of senescent cells to secrete proinflammatory cytokines [235]. These molecules then contribute to inflammaging by perpetuating and amplifying inflammation and maintaining an activation state of innate immune cells. In evaluating the diverse data examining the role of microglia and neuroinflammation in neurodegenerative disease, one must take into consideration all of the comorbidity features, including those reflected in the parallel adaptive/remodeling processes associated with inflammaging and immunosenescence.

## 7. Summary

The overwhelming evidence is that microglia clearly play a contributory role in neurodegeneration. Whether or not they initiate a neurodegenerative process is more difficult to conclude. Each individual function of microglia contributes to maintaining a homeostatic balance within the brain, and the complexity of the response occurring during the shifting stages of any insult or injury requires consideration [236,237,238]. The homeostatic balance of the brain relies on a complex communication between individual cells. This interaction ensures that altered neuronal activity, and likely neuroglia activity, will be communicated rapidly to microglia. Any event that would modify this balance could be detrimental. While the production of pro-inflammatory cytokines is often considered to lead to an adverse environment for neurons, what may be of more concern is a compromise of normal microglial functions. The inability to mount an appropriate pro- or anti-inflammatory response, detect and migrate to site of injury, or perform phagocytosis and degrade debris or aberrant proteins, would be determining factors in brain injury and repair. Within this spectrum of biological processes and functions, microglia represent an influential target for alterations from environmental exposures that could have significant influence on the time of onset or progression of a neurodegenerative disease.

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
