# Peer review of "Microglia in Neurodegenerative Events—An Initiator or a Significant Other?"

_ijms, 2021, doi:10.3390/ijms22115818_

Round 1

Reviewer 1 Report

The present article profoundly reviews the ongoing knowledge about microglia implication in neurodegeneration, summarising many molecular mechanisms and cellular pathways with exquisite detail. I have some suggestions and comments that could be employed to improve the review:

  • The article is well written and scientificaly consistent, but in general I would reccomend to review the structure of the text in order to build a history around the topic insetead of listing many individual data, making the reading easier. For example: why are Parkinson’s disease and stroke the only pathological processes considered? Why is aging taking into account apart of them?
  • In line 39 the author refers to the infiltration of peripheral immune cells and the topic arise again speaking about stroke at the end of the review. However, it would be necessary to mention that in neurodegenerative or brain injury diseases in which the stability of the blood brain barrier becomes compromised, microglia can be easily mixed with and mistaken for peripheral macrophages. I strongly recommend mentioning this possibility since it sometimes challenges the methodological approaches described along the review.
  • In line 100 the author refers to the neuroprotective or neurodegenerative phenotype that can be shown by microglia according to the canonical in vitro model of M0-M1-M2 activation states. However, it is currently accepted that in vivo microglial phenotype is much more variable and does not follow that dichotomy. It would be interesting to mention this phenotype variability. A work that might be useful here is: Gray, S. C., Kinghorn, K. J., & Woodling, N. S. (2020). Shifting equilibriums in Alzheimer’s disease: the complex roles of microglia in neuroinflammation, neuronal survival and neurogenesis. Neural regeneration research, 15(7), 1208
  • It would be also important to mention the controversy around the detrimental or beneficial role of microglia activation in some neurodegenerative or brain injury diseases depending on the temporal point of the disease. For example, in traumatic brain injury: Chiu, Chong-Chi, et al. "Neuroinflammation in animal models of traumatic brain injury." Journal of neuroscience methods 272 (2016): 38-49.
  • A proper reference should be included for the statement in lines 117-120
  • In line 194 “down-steam” should be probably replaced by “downstream”
  • The consistency of acronyms and their abbreviated terms should be reviewed along the text. For example, in line 305 the author refers to “NHD brains” and the acronym NHD has not been previously explained and, in the contrary, in line 310 or 582 the terms “tumour necrosis factor-α” or “interleukin 1-β” are used again although they have been previously presented as TNF-α or IL-1β
  • A proper reference should be included for the statement in lines 406-408
  • References should be reviewed along the text in order to avoid duplications. For example, reference nº 42 and nº 100 refer to the same article

Author Response

I thank the reviewer for excellent suggestions to improve the manuscript.

References have been added as requested

duplicate citations removed

acronyms and order of presentation has been corrected and checked

The recommendation to provide a background framework for some of the sections has been addressed by the addition of such at the beginning of each major section.  I agree with the reviewer that this would be very helpful to the reader

For the section on the diseases I have provided a rationale for looking at stroke as an acute damage and PD as an example of progressive degeneration.  I moved aging section to follow PD and the consideration of aging as a significant comorbidity that must be considered when evaluating microglia responses in age-related neurodegenerative disorders.  I think this provides a much better flow.  

The influence of infiltrating cells, the inability to distinguish resident microglia from blood borne macrophages is mentioned and the effects on the BBB with disease are not mentioned.

I have also tried to identify the concern for trying to identify the detrimental vs beneficial effects of microglia within the nature of their normal function within the paper.  

I appreciate the point raised regarding microglia polarization and agree that this is a major issue as it is often overlooked.  This was why I originally refrained from using terms such as M1/M2 and rather used the functional aspect of the response.  I have now included a section on this with information on the nature of the diversity of the profiles of macrophages depending on the initiating stimuli.  

I thank the reviewer for the recommended citations.

All additions are provided in the track-changes version of the paper as they are somewhat lengthy to include within this response.  

Reviewer 2 Report

This review extensively goes over what is currently known about the role played by microglia and further add a significant discussion a hypothetical function of this glial cell as a potential initiator of neurodegeneration.  The author did an outstanding job and is to be commended on the thoroughness of the review. This review includes updated information with a huge number of references. This article remains complete and timely, and is recommended for publication.

I suggest a few, minor points that could be added in the final version of the manuscript:

1) Although the author slightly mention it, the effects of an anisomorphic microglia reaction versus isomorphic as a “double-edge sword” mechanism of the microglia (Wyss-Coray and Mucke, 2002; Neuron) could be mentioned in more detail (maybe as a subsection). In this regard, neurodegenerative processes affecting different neurons with the same mutation result in significantly different microglia-mediated responses (Baltanas et al, 2013; Glia).  

2) A graphical abstract showing a summary of the information that the author attempt to show would help to the reader about the significance of the review.

Author Response

I thank the reviewer for the helpful comments

I have tried to develop the idea of multiple effects of microglia a bit better.  I have stayed away from the specific terminology of anisomorphic versus isomorphic simply because it is not commonly used even though an interesting concept in 2002.  I tried to incorporate briefly that the response depends on the nature of the damage/insult and the microenvironment.  I found the Baltanas paper of interest and it did support the idea that microglia respond differently depending on the nature of the neuronal response and the severity or stage of that response.  

A graphical abstract could be of interest but I found when I tried to develop this for the initial submission and then again now that to capture the points in the review that it became somewhat cumbersome. While covering many processes it could not be fully complete.   I selected review articles on specific topics that I felt did provide a good graphic representation of that specific aspect that I hope will be of help to the reader.  

Round 2

Reviewer 1 Report

The changes made are considerable and the review has improved to the point to be better published. It is now even more interesting and understable that at the beginning